# Combustible ice mimicking behavior of hydrogen-bonded organic framework at ambient condition

Yang Wang[1], Xudong Hou[1], Congyan Liu[1], Mohamed K. Albolkany [1], Yan Wang[1], Niannian Wu[1], Chunhui Chen[1] & Bo Liu [1✉]

Adsorption of guest molecules by porous materials proceeds in a spontaneous exothermic way, whereas desorption usually requires external energy input as an endothermic process. Reducing such energy consumption makes great sense in practice. Here we report the reversible and automatic methanol (MeOH) adsorption/release in an ionic hydrogen-bonded organic framework (iHOF) constructed from guanidinium cation and borate anion ([B $(OCH_3)_4]_3[C(NH_2)_3]_4Cl\bullet4CH_3OH$, termed Gd-B) at ambient condition. The metastable Gd-B automatically releases all sixteen MeOH molecules (63.4 wt%) via desorption and tetramethyl borate hydrolysis at ambient atmosphere and the structure can be recovered when re-exposed to MeOH vapor or liquid, mimicking combustible ice behavior but at ambient condition. Reversible capture/release of four guest MeOH molecules is also realized without destroying its crystal structure. The combustible Gd-B paves a way for exploring metastable iHOF materials as carrier for alternative energy source and drug delivery etc.

[1] Hefei National Laboratory for Physical Sciences at the Microscale, Fujian Institute of Innovation of Chinese Academy of Sciences, School of Chemistry and Materials Science, University of Science and Technology of China, Hefei, Anhui 230026, China. ✉email: liuchem@ustc.edu.cn

Natural gas hydrate (NGH) or combustible ice, a potentially alternative energy source in place of conventional coal and oil, is receiving growing attention to address the existing energy crisis[1,2]. In NGH, gas molecules (primarily methane) are trapped in solid water with a cage crystal structure under low temperature and high pressure[3–5], which is prone to collapse and release the trapped gas component under increasing temperature and/or reduced pressure (Fig. 1a). The recharge of methane in water that denotes the formation of NGH, is desirable but challenging as harsh condition are of necessity (<10 °C and >30 atm)[6,7]. Nevertheless, it is highly revelatory for us to mimic this process for storing energy-containing materials, via molecule inclusion during host formation and release when host breaking in mild condition, for example, ambient atmosphere. For cycling usage, the broken host materials should revert with recharge of released guest molecules in a facial and economic way, which is impracticable for NGH.

A diverse class of porous materials, including carbon, zeolites, metal organic frameworks (MOFs), covalent organic frameworks (COFs), and porous molecular crystals etc., have been developed to function as sorbents for gas storage owing to their tunable structures, pores sizes, and high surface area[8–12]. Although they exhibited excellent gas-storage capability, the storage/release mechanism is totally distinct from the process in NGH (Fig. 1). Specifically, these porous materials adsorb guest molecules by virtue of high surface area and thus strong surface affinity, whereas release of guest molecules, especially for volatile organic compounds, often needs extra energy (heating and/or vacuuming), as shown in Fig. 1b. In contrast, the methane molecules release from NGH is furious and thus out of control at ambient condition.

Although a variety of strategies on enhancing the adsorptive capacity have been extensively proposed, facial control on release of adsorbed molecules is little explored. In order to achieve controllable and reversible adsorption/release of guest molecules, there are two possible pathways: either adjusting the guest–host interaction or controlling the reversible framework transformation based on guest release and capture. The synergy between the two pathways can provide better control over the ad/desorption behavior. Some recent studies focused on the control of the guest–host interactions. For example, Kim et al. reported a Zr–MOF-based device to achieve water desorption from MOF with a low solar energy input of 1 kW m$^{-2}$[12]. Cadiau et al. developed a fluorinated MOF with a periodic array of open metal coordination sites and fluorine moieties within the one-dimensional channel, which releases adsorbed water at relatively moderate temperature (~105 °C), about half the energy input for commonly used desiccants[13]. In terms of MOF and COF studies, intensive attention has been paid to improve their stability. Nevertheless, most porous frameworks collapsing by thermodynamic and/or chemical ways cannot be recovered, which renders the second strategy inaccessible.

Hydrogen-bond (H-bond) assisted supermolecular assembly shows great structural flexibility owing to the moderate strength of H-bonds, which could be ideal candidate for controllable capture/release of guest molecules at mild condition[14–17]. The past decades have witnessed the emerging progress of hydrogen-bonded organic frameworks (HOFs) constructed via intermolecular H-bonds[18–23] H-bond originated from electrostatic attraction is readily formed among adjacent organic molecules bearing electronegative groups, which does not ask for extra energy for reaction[24]. Therefore, HOFs can be prepared via dissolving and recrystallizing of organic molecules, while the weak noncovalent interaction of H-bond renders most HOFs fragile, getting collapsed upon removal of guest molecules[18]. Only few HOFs show permanent porosity for adsorption[24–26].

In addition to the reported HOFs comprising neutral organic molecules[18], another type of HOFs is constructed from cations and anions. Herein, these charge-assisted frameworks are denoted as ionic HOFs (iHOFs), in which electrostatic attraction between cations and anions plays a vital role for iHOF assembling to strengthen the stability of established framework, beyond H-bonding and Van der Waals force[17,27,28] Guanidinium cation, coupled with sulfonate $[SO_4]^{2-}$, has been adopted to construct iHOFs[29–31]. In 2005, Abrahams et al. reported the synthesis and structural characterization of iHOF $[B(OCH_3)_4]_3[C$

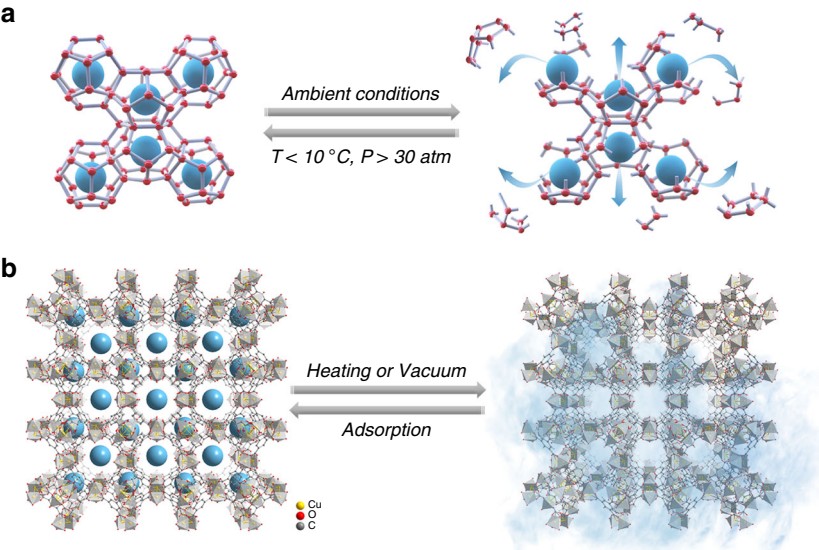

**Fig. 1 Guest capture/release mechanism. a** Methane molecules (blue balls) stored within water cage (constructed by red balls and gray sticks) and released (denoted by blue arrows) after cage collapse under different condition, the typical hydrate structure I comprising large $5^{12}6^2$ (twelve pentagonal and two hexagonal faces) and small $5^{12}$ cages is taken for illustration. **b** Methane molecules (blue balls) stored in MOF (HKUST-1 as an example) cavities release upon applying external energy (heating or vacuum). Orange, red, gray balls, and light gray tetrahedrons denote Cu, O, C atoms, and the subunits within HKUST-1, respectively. H atoms are omitted for clarity. The blue smog denotes the methane gas atmosphere.

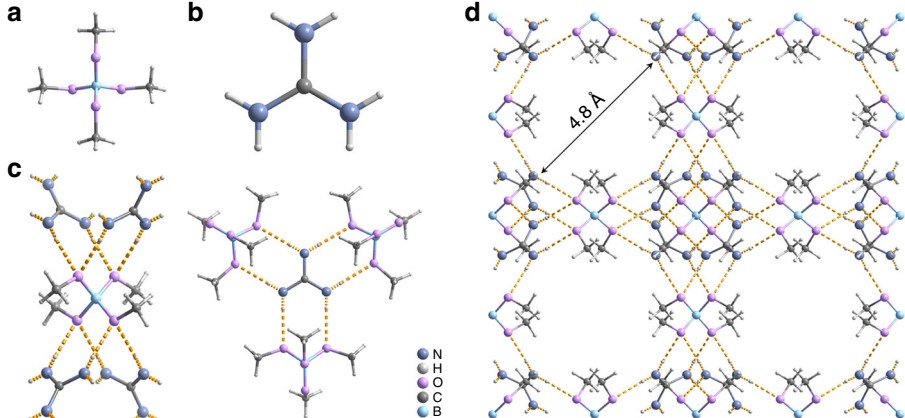

**Fig. 2 Crystal structure of Gd-B. a** Borate ester anion. **b** Guanidinium cation. **c** H-bond linking (denoted as dashed yellow line) between **a** and **b**. **d** Gd-B framework comprising borate ester anion and guanidinium cation via H-bond linking (highlighted by dashed yellow line, disordered Cl⁻ and guest MeOH molecules are omitted for clarity). The Gd-B framework features a pore size of 4.8 Å that enables the inclusion of MeOH molecules. Grayish blue, light gray, pink, gray, and pale blue balls denote N, H, O, C, and B atoms, respectively.

$(NH_2)_3]_4Cl\cdot4CH_3OH$ (termed Gd-B)[31]. In this work, we explore the reversible structural transformation of Gb-B upon MeOH capture and release without extra energy input, which mimics NGH in terms of the adsorption-release behavior of guest molecules while differs in operation conditions (ambient atmosphere vs. harsh condition) and adsorbates (methanol vs. methane). Similar with combustible ice, we obtain combustible Gd-B, in which MeOH can be directly released into air for lighting at ambient condition.

## Results

**General information and XRD analyses of Gd-B**. Gd-B was prepared according to the reported procedures ("Methods" section and Supplementary Table 1)[31]. The borate ester anion and guanidinium cation (Fig. 2a, b) are assembled via electrostatic interaction and H-bond into framework. Each borate ester anion is connected with four guanidinium cations via H-bond, while each guanidinium cation bridges three borate ester anions (N–H–O, N–O distance: 2.914 Å, angle: 174.68°) (Fig. 2c). The spatial extension of these (3,4)-connected units renders a three-dimensional H-bonded framework ((63)(6284) or boracite topology) with pore size of 4.8 Å (Fig. 2d). As shown in Fig. 3a, the PXRD patterns indicate that the crystal structure of as-prepared Gd-B keeps good consistency with that of simulated from single-crystal XRD data. And elemental analysis (Supplementary Table 2) reveals that fresh Gd-B matches well with the formula $[B(OCH_3)_4]_3[C(NH_2)_3]_4Cl\cdot4CH_3OH$. We found that transparent regular tetrahedron of freshly prepared Gd-B turned into white powder when exposing in air (Supplementary Fig. 1). Specifically, the fresh Gd-B gradually lost its own crystallinity upon exposure in air, i.e., for 6 h (Supplementary Fig. 2). Prolonged exposure led to structural transformation and produced a new phase (for example, air drying for 24 h) totally different from fresh Gd-B (Fig. 3a). We did not observe further phase transformation by exposure in air up to 72 h, indicating the favorable stability of the transformed powder phase (Supplementary Fig. 2). Importantly, when this powder (termed air-dried Gd-B) was exposed to MeOH vapor, for example 24 h, the structure of Gd-B can be recovered (Fig. 3a). This reversible structural transformation can also be realized by directly dissolving the powder in liquid MeOH and recrystallizing via MeOH evaporation (Supplementary Fig. 3). All these processes took place at ambient condition. Also, the air-dried Gd-B can be readily dissolved in $H_2O$ and crystallized into guanidinium tetraborate dihydrate

$([C(NH_2)_3]_2[B_4O_5(OH)_4]\cdot2H_2O$, Supplementary Fig. 4)[32], which was stable in air atmosphere but turned into Gd-B again when dissolving and recrystallizing in MeOH (Supplementary Methods and Supplementary Fig. 5). These findings illustrate that Gd-B undergoes reversible structure transformation upon release and re-adsorption of MeOH. Furthermore, the H-bonded framework of Gd-B can be maintained upon vacuum drying of both fresh Gd-B (Supplementary Fig. 6) and recovered Gd-B (Fig. 3a).

**NMR analyses of Gd-B**. We further carried out the $^1H$-NMR and $^{13}C$-NMR analyses over Gd-B samples. As shown in Fig. 3b, fresh Gd-B features two peaks at 3.17 ppm and 4.04 ppm, which are assigned to hydrogen from the $-CH_3$ and $-OH$ groups in MeOH, respectively. Note that the detection of $-NH_2$ linked groups within the structure is not available using $^1H$-NMR analysis, owing to the existence of active protons that may drift over a wide range of chemical shifts[33]. After vacuum drying for 24 h, in spite of removing MeOH guest molecules, the same $^1H$ chemical shifts are evidenced from the $-OCH_3$ group in borate ester (Fig. 2a). Further $^{13}C$-NMR tests (Fig. 3c) confirm the existence of the ester group $-OCH_3$, as the peak at 48.59 ppm can be detected even after vacuum drying over fresh Gd-B for 1 h. However, this peak indicative of $-OCH_3$ is absent in air-dried Gd-B samples, suggesting the decomposition of the framework upon interaction with moisture in ambient atmosphere. When the air-dried Gd-B was exposed to MeOH vapor for 24 h, reversible structural transformation was observed, as indicated by the peak at 48.59 ppm, which is identical with that of fresh Gd-B. As for the $^1H$-NMR spectra of air-dried Gd-B samples, the chemical shift at ~3.17 ppm gets continuously weakened and finally disappears with prolonging exposure in air for drying (Fig. 3d). All these results match well with XRD analyses.

**Structural transformation of Gd-B**. Combined with Gd-B crystal structure, XRD, and NMR analyses, we could come to a convincing interpretation of the structure transformation during its exposure in air and structural restoration as displayed in Fig. 4. Fresh Gd-B firstly loses the accommodated 4 MeOH molecules at the beginning of exposure in air; subsequently, borate ester anion of $[B(OCH_3)_4]^-$ gets hydrolyzed by moisture and further releases 12 MeOH molecules (Supplementary Fig. 7). Upon exposure to MeOH, the residual white powder (air-dried Gd-B) experiences the formation of borate ester to re-build the Gd-B framework and adsorption of MeOH to fill in the cavity, giving rise to the

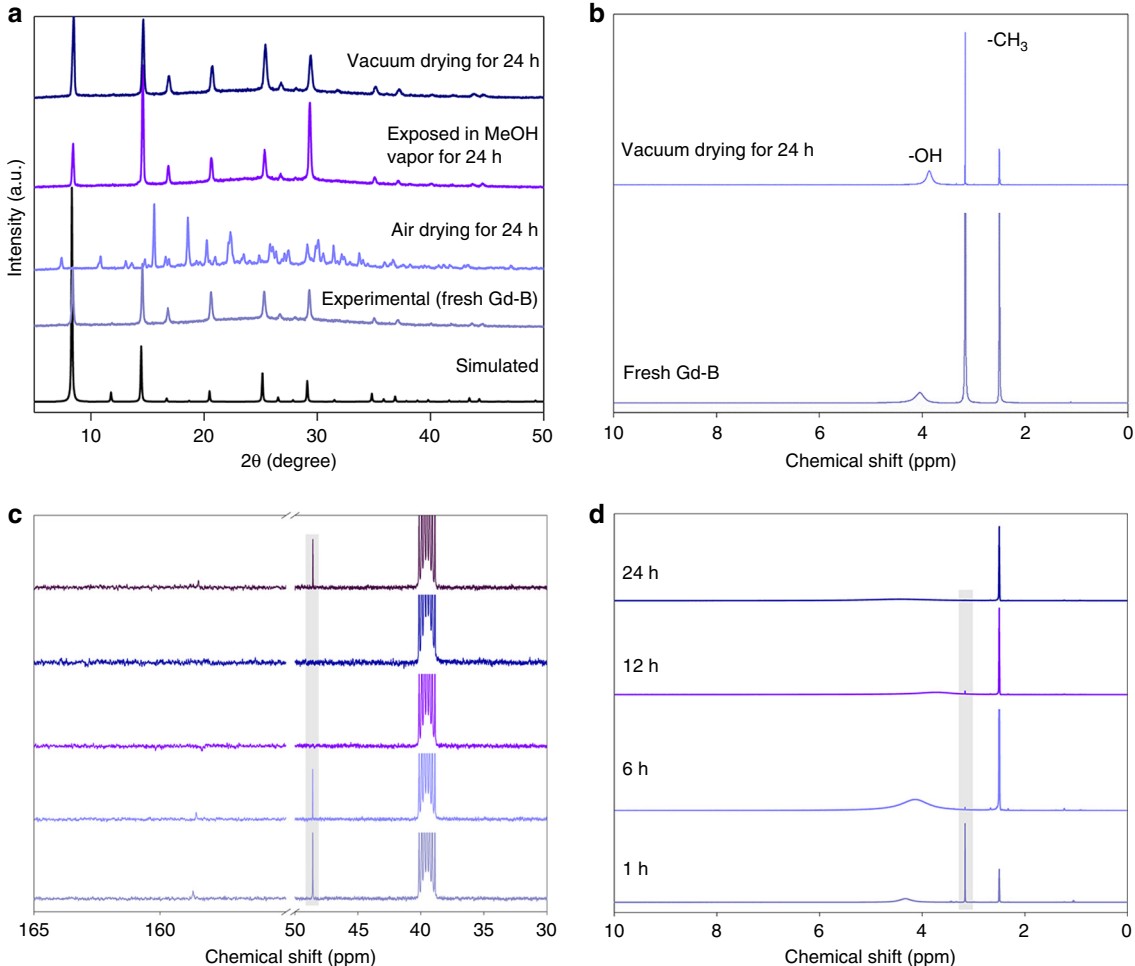

**Fig. 3 Characterizations of structural evolution of Gd-B. a** Powder X-ray diffraction (PXRD) patterns of fresh Gd-B underwent successive treatments under different conditions. Specifically, fresh Gd-B was firstly exposed in air atmosphere for 24 h, the dried powder sample was then exposed in MeOH vapor for 24 h, followed by vacuum drying for 24 h. **b** $^1$H-NMR spectra of fresh Gd-B and Gd-B dried under vacuum for 24 h. **c** $^{13}$C-NMR spectra of fresh Gd-B, Gd-B dried under vacuum for 1 h, air-dried Gd-B and then vacuum dried for 1 h, air-dried Gd-B, and then exposed in MeOH vapor for 24 h (from bottom to top). **d** $^1$H-NMR spectra of fresh Gd-B under air drying for different time. d$_6$-DMSO was utilized for these characterizations.

restoration of Gd-B crystal structure (path 1 in Fig. 4). Alternatively, the restoration process can be facilitated by two-step recrystallization, benefiting from the facial transformation among $[B(OH)_4]$, $[B_4O_5(OH)_4]$ (Supplementary Fig. 4) and $[B(OCH_3)_4]$ groups (path 2 in Fig. 4). It is also evidenced that vacuum treatment can only remove the MeOH existing as a guest molecule and Gd-B framework structure remains intact (Supplementary Fig. 6). We experimentally confirm the Gd-B structure collapse and reconstruction with MeOH release and capture, which mimics the NGH behavior for energetic molecules storage, but under mild condition without extra energy input.

**Adsorption and release behavior of Gd-B.** As MeOH in Gd-B HOF can be readily released in air, we directly activated fresh Gd-B sample via vacuum at room temperature for nitrogen isotherm measurement. Nitrogen ad/desorption at 77 K reveals a typical type-I isotherm with a specific surface area of 257 m$^2$ g$^{-1}$ (Fig. 5a), and Gd-B exposed to N$_2$ atmosphere at varied pressure during the test signifies its stability in inert gas, where structural transformation into air-dried Gd-B will not take place. The pore size distribution indicates a primary pore size of ~4.85 Å (Supplementary Fig. 8), which is in line with the structural analysis in Fig. 2d. The H$_2$ adsorption amount at 77 K is determined to

be ca. 50 cm$^3$ g$^{-1}$. The sample exhibits a CO$_2$ adsorption of ca. 13 cm$^3$ g$^{-1}$ at 298 K. The obvious hysteresis is ascribed to the strong interaction between acidic CO$_2$ and the amino group of guanidinium cation in Gd-B; similar hysteresis has also been reported in amino-functionalized MOFs that enhance CO$_2$ adsorption[34,35].

Encouraged by the reversible structure collapse and restoration of Gd-B upon release and adsorption of MeOH at ambient condition without extra energy input, MeOH sorption behaviors are evaluated in more details to reveal these processes and provide guidance for rational design of functional materials as carriers for alternative energy source. Specifically, we collected MeOH-sorption isotherms over nonporous air-dried Gd-B (Supplementary Fig. 9) at 298 K for multiple runs (Fig. 5b), where after each run, the sample is online evacuated for activation without exposure to air. For the first run, MeOH adsorption over air-dried Gd-B reaches up to 417 cm$^3$ g$^{-1}$, corresponding to 59.6 wt% of MeOH in fresh Gd-B that contains 16 methoxyl groups. The MeOH adsorption capability is high among the reported adsorbents, which is comparable with MIL-100 having specific area >3000 m$^2$ g$^{-1}$ (Supplementary Table 3). The result shows little deviation from theoretically calculated value (63.4 wt%). It is noted that at initial stage, there is no MeOH adsorption occurring till the P/P$_0$ of 0.45. Subsequently, MeOH adsorption increases

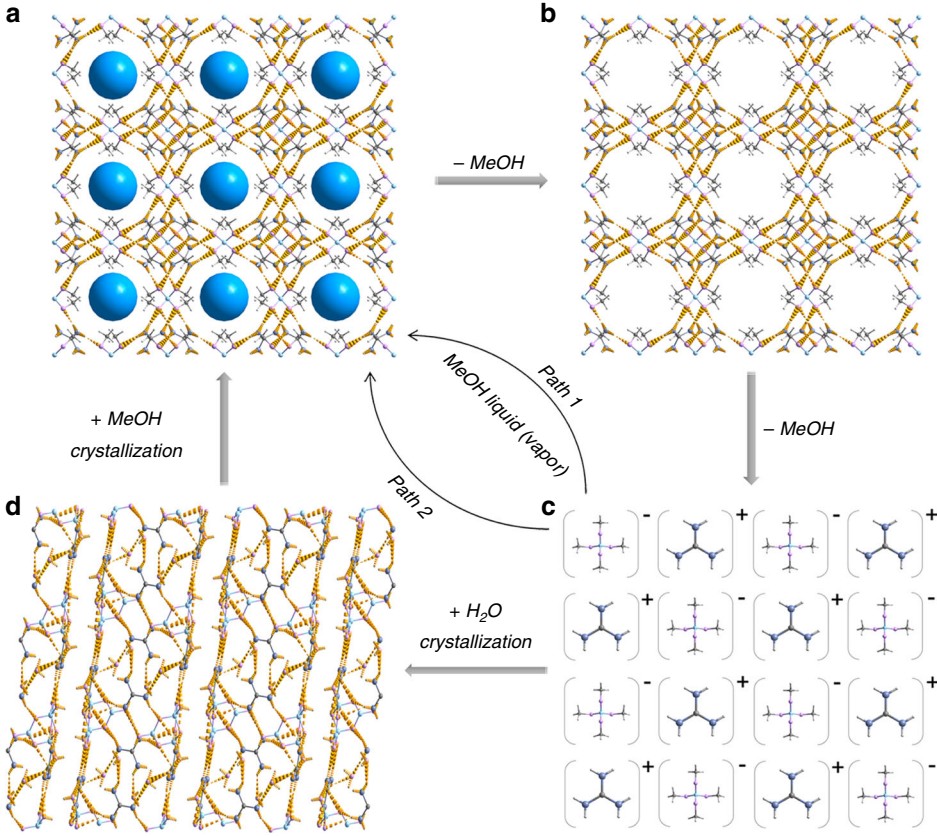

**Fig. 4 Structural transformation of Gd-B samples at ambient condition. a–c** Fresh Gd-B (**a**) loses accommodated MeOH (blue balls) molecules (**b**) upon exposure in air atmosphere, prolonged exposure results in the hydrolysis of borate ester anion of $[B(OCH_3)_4]^-$ by moisture, which further releases twelve MeOH molecules and turns into white powder (**c**, named as air-dried Gd-B). Grayish blue, light gray, pink, gray, and pale blue balls denote N, H, O, C, and B atoms, respectively. The dashed yellow lines denote the H-bond linking. **d** Crystal structure of $[C(NH_2)_3]_2[B_4O_5(OH)_4]\bullet2H_2O$. Two paths are available for the structural restoration of Gd-B. Typically, air-dried Gd-B is exposed in MeOH liquid or vapor to obtain Gd-B (path 1); air-dried Gd-B is dissolved in $H_2O$ and recrystallized into $[C(NH_2)_3]_2[B_4O_5(OH)_4]\bullet2H_2O$, which is further dissolved and recrystallized in MeOH to obtain Gd-B (path 2).

steeply, indicating a breakthrough point. The adsorption amount of ~117 $cm^3\,g^{-1}$ (0.167 $g\,g^{-1}$) at breakthrough point that accounts for about 1/4 of the total adsorption amount, corresponding to physically adsorbed four MeOH molecules in the formula of $[B(OCH_3)_4]_3[C(NH_2)_3]_4Cl\bullet4CH_3OH$. The remaining 3/4 adsorption amount of ~300 $cm^3\,g^{-1}$ (0.429 $g\,g^{-1}$) is identified at higher relative pressure, corresponding to 12 methoxyl groups on borate ester in Gd-B. However, we realized that physical adsorption cannot be reached prior to Gd-B framework restoration. We carried out the 2nd run ad/desorption test after vacuum treatment over the air-dried Gd-B. The isotherm indicates that the breakthrough point disappears and the total adsorption amount is decreased to be ca. ~300 $cm^3\,g^{-1}$. Subsequent 3rd and 4th runs of ad/desorption also show continuous decrease of MeOH adsorption amount (Fig. 5b). The MeOH sorption behavior of the 5th run is almost identical with that in 4th run (Supplementary Fig. 10). Meanwhile, fresh Gb-B treated by vacuum exhibits the almost overlapped sorption isotherm with that of air-dried sample at 5th run (Fig. 5c), with a saturated adsorption amount of 118 $cm^3\,g^{-1}$ and 116 $cm^3\,g^{-1}$, respectively. This is in line with physically adsorbed four MeOH molecules in Gd-B.

As demonstrated by the XRD results (Fig. 3a), the complete recovery of Gd-B structure from air-dried Gd-B sample in MeOH vapor is a kinetic process that requires a period of hours. This is also reflected in the continuous MeOH sorption tests in Fig. 5b, where only part of adsorbed MeOH participates in Gd-B reconstruction in each run of sorption owing to kinetic factor.

The MeOH adsorption ratio of 1:3 in air-dried Gd-B sample in 1st run (Fig. 5b) can be explained by the partial formation of borate ester at breakthrough point, rather than physical adsorption prior to Gd-B framework formation. Under vacuum treatment, borate ester is stable so that the adsorption amount of subsequent runs continuously reduces till all borate ester groups have been re-generated (after four runs). The results indicate that the restoration of air-dried Gd-B to Gd-B follows a stepwise way as demonstrated in Supplementary Fig. 11. Our combined findings on sorption tests over both air-dried and fresh Gd-B further support the reversible structure transformation of Gd-B. The cycling performance of MeOH adsorption–desorption of fresh Gd-B was also evaluated (Fig. 5d), which suggests no apparent adsorption capability loss for cycling usage.

It is well documented that a diverse class of MOFs and other nanoporous materials exhibit much higher MeOH adsorption amount, while high temperature and reduced pressure are of necessity to release MeOH from the parent framework[36–41]. In sharp contrast, MeOH adsorbed in iHOF of Gd-B can automatically release MeOH at ambient condition. Interestingly, the accommodated MeOH molecules in Gd-B can be directly released at ambient condition for lighting (Supplementary Movie 1) without structural collapse (Supplementary Fig. 12). It is also verified by the example that HKUST-1-adsorbed MeOH cannot be lighted in air, though HKUST-1 can adsorb much more MeOH, because very little MeOH can be released from HKUST-1 at ambient condition.

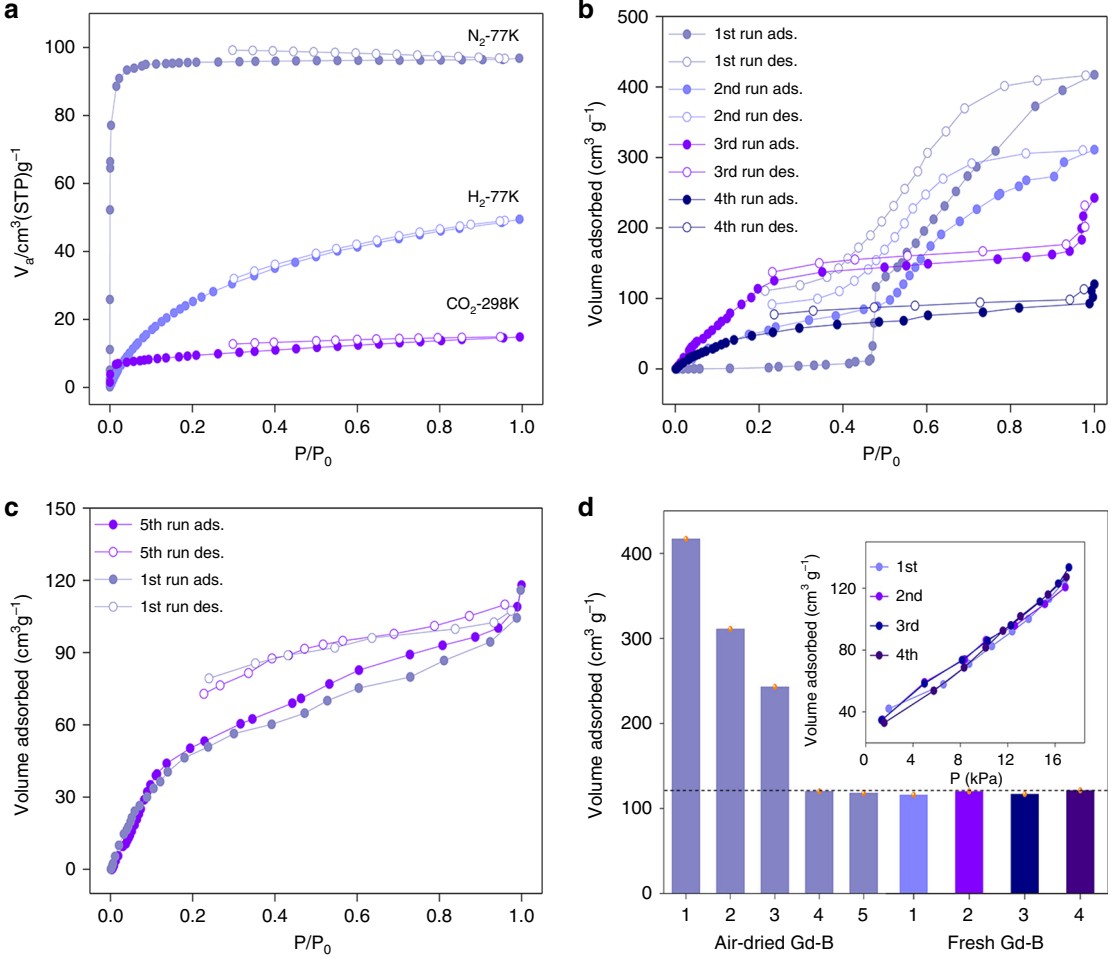

**Fig. 5 Sorption behavior of fresh and air-dried Gd-B samples. a** $N_2$ (77 K), $H_2$ (77 K), and $CO_2$ (298 K) sorption isotherms of fresh Gd-B activated by vacuum at room temperature. **b** MeOH sorption isotherms over air-dried Gd-B at 298 K for different runs. For each run, the sample is online activated by vacuum without exposure to air. **c** The fifth run of MeOH sorption isotherms (violet) over air-dried Gd-B at 298 K, together with the first run of MeOH sorption isotherms (gray) over fresh Gd-B treated with vacuum. **d** Cycling MeOH sorption performance of air-dried Gd-B and fresh Gd-B, the sorption values are denoted by orange dots. Inset: single-point MeOH adsorption test of fresh Gd-B, the target pressure was set as 16 kPa according to the MeOH vapor pressure at 298 K.

## Discussion

Our findings reveal that metastable Gd-B is an excellent MeOH carrier as it captures and releases MeOH that accounts for about 60% of Gd-B weight, based on its automatic and reversible structural transformation at ambient condition. Intensive studies on porous frameworks have been focused and advanced on their chemical, thermal, and mechanical stability for the long-term durability towards certain applications. Relatively weak intermolecular interactions like H-bond, Van der Waals, and electrostatic forces usually lead to metastable frameworks, which are considered to be a disadvantage. However, release of the volatile organic compound (VOC) and then reuse of adsorbents require extra energy input to eliminate the guest–host interactions. Reversible structure transformation upon guest molecules adsorption and release under mild condition greatly benefits energetic material unitization, moisture capture/release. Such metastable frameworks are also potential candidates for drug delivery.

## Methods

**Chemicals and materials.** Guanidine hydrochloride ($CN_3H_5 \cdot HCl$, AR, ≥99.5%) and boric acid ($H_3BO_3$, AR, 99%) were purchased from Shanghai Macklin Biochemical Co. Ltd. (China). Methanol (MeOH, AR, ≥99.5%) and triethylamine

(TEA, AR, ≥99%) were purchased from Sinopharm Chemical Reagent Co., Ltd. (China). Dimethyl sulfoxide-$d_6$ was purchased from Adamas-Beta.

**Characterization.** Powder X-ray diffraction (PXRD) tests were carried out on a Rigaku MiniFlex 600 X-ray diffractometer using Cu Kα radiation (λ = 1.54178 Å). Elemental analyses (EA) were completed with a Vario EL III Elemental Analyzer (Elementar Inc.). The [1]H-NMR and [13]C-NMR spectra were recorded on a Bruker AVANCE AV III 400WB spectrometer operating at 400 MHz. Gas-sorption isotherms were measured at 77 K or 298 K, and methanol vapor sorption isotherms were measured at 298 K on a BEL sorp-max machine, BEL, Japan.

**Preparation of Gd-B single crystal.** Gd-B was synthesized according to previously reported procedures (CCDC number: 1686051)[31]. Briefly, 1.86 g $CN_3H_5 \cdot HCl$ was dissolved in 21 mL MeOH to form colorless solution, which was further mixed with the solution prepared by dissolving 0.6 g $H_3BO_3$ into 24 mL MeOH that contained 1.35 mL TEA. The resultant mixed solution was subjected to free standing for 12 h in an open vial, upon which large amount of colorless tetrahedral crystal can be obtained at room temperature. The sample was further washed with MeOH quickly and then collected by centrifugation at 10,000 rpm/min for 3 min. The sample is denoted as Gd-B.

**Reversible structure transformation of Gd-B.** The freshly prepared Gd-B single crystal turned into wet white powder when exposed in air. The Gd-B powder (50 mg) stored in the tube (5 mL) was placed into a 100-mL beaker which contained 30 mL of MeOH and covered by sealing film, upon which an artificial MeOH vapor atmosphere was created. The powder was exposed in the vapor for 24 h to complete the recovery process of Gd-B structure. An alternative way to

realize the reversible structural transformation can be realized by directly dissolving 20 mg of white powder into 0.7 mL of MeOH solvent. The tetrahedral colorless single crystals were obtained later after dissolution and recrystallization for 12 h.

Being similar with the recrystallization in MeOH, 20 mg air-dried Gd-B was dissolved in 0.7 mL of $H_2O$ and subjected to recrystallization for 72 h under air atmosphere. This process resulted in the formation of colorless single crystal with different structure from Gd-B, which was a known structure reported by T. J. R. Weakley and named as guanidinium tetraborate dihydrate ($[CN_3H_6]_2[B_4O_5(OH)_4]\cdot 2H_2O$, Supplementary Fig. 4, CCDC number: 1132636)[32]. Subsequently, 20 mg of the crystal, which was stable in air atmosphere, was grinded into white powder and dissolved in 1 mL of MeOH by gentle sonication for several minutes. The solution was then subjected to recrystallization for 24 h under air atmosphere, upon which Gd-B single crystal can be obtained.

**MeOH sorption over air-dried and fresh Gd-B**. The fresh Gd-B crystal was exposed in air for 24 h to transform into the powder form with no further structural change. Next, the powder (air-dried Gd-B) was subjected to vacuum for 12 h ($10^{-7}$ Pa) at 60 °C prior to MeOH sorption test at 298 K. To confirm the structural collapse and recovery, the air-dried Gd-B was directly subjected to multiple-run MeOH sorption test at 298 K. For each run, the sample was online-activated on the sorption machine by vacuum without air interference.

The fresh Gd-B, $[B(OCH_3)_4]_3[C(NH_2)_3]_4Cl\cdot 4CH_3OH$, was vacuumed for 24 h ($10^{-7}$ Pa) at room temperature to initiate MeOH sorption test at 298 K. The sorption result of fresh Gd-B was compared with the fifth-run sorption of air-dried Gd-B, which had completed structural recovery through four runs of MeOH ad/desorption tests.

**Automatic release of four guest MeOH molecules from bulky Gd-B**. Typically, fresh Gd-B stored in MeOH was collected and put onto filter paper, upon which the surface attached MeOH on Gd-B were removed. Subsequently, the fresh Gd-B crystal (500 mg) was placed into a silica tube (length: 15 cm, inner diameter: 10 mm) and covered by sealing film. The tube was left untouched for 30 min to be fully filled with released MeOH guest molecules. A cigarette lighter was then placed at the tube outlet to ignite the released guest MeOH molecules from bulky Gd-B, the flame can last for few seconds (Supplementary Movie 1). After that, the sample was weighed to evaluate the mass change before and after ignition. Also, the crystal structure of Gd-B after ignition was examined by XRD.

## Data availability

All data generated or analyzed during this study are included in this article and its Supplementary Information files, other data that support the findings of this study are available from the corresponding author upon request. The X-ray crystallographic coordinates for structures reported in this study have been deposited at the Cambridge Crystallographic Data Centre (CCDC), under deposition numbers 1686051 (Gd-B) and 1132636 ($[C(NH_2)_3]_2[B_4O_5(OH)_4]\cdot 2H_2O$). These data can be obtained free of charge from The Cambridge Crystallographic Data Centre via www.ccdc.cam.ac.uk/data_request/cif.

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

## Acknowledgements

We acknowledge support from Hefei National Laboratory for Physical Sciences at the Microscale, Hefei Science Center of Chinese Academy of Sciences, Fujian Institute of Innovation of Chinese Academy of Sciences, the National Natural Science Foundation of China (NSFC, 21571167, 51502282), the Fundamental Research Funds for the Central Universities (WK2060190053 and WK2060190100) and Anhui Province Natural Science Foundation (1608085MB28).

## Author contributions

B.L. conceived and designed the experiments. Yang W. performed the experiments and analyzed the data. X.H. helped in the adsorption test of the samples. C.L., M.A., and Yan W. conducted parts of the mechanism analyses. N.W. helped in the NMR tests and analyses. C.C. analyzed the X-ray crystal structure of the single crystals. B.L. and Yang W. co-wrote the paper with input from all authors. All authors discussed the results and commented on the paper.

## Competing interests

The authors declare no competing interests.
