## [Peer Review File · Nature Communications]

Reviewers' comments:

Reviewer #1 (Remarks to the Author):

This manuscript is far below the standards of Nature Communications. The behavior described is neither novel, unique, or interesting. There is a growing trend to use the term HOF. The authors here have now expanded on this to iHOF, i meaning ionic. Some authors (but not these) have taken to numbering them, like HOF-10, HOF-11, ignoring hundreds of "HOFs" reported long before. The use of this acronym, in the view of this reviewer, is an unsavory practice and it ignores a lot of history.

This article essentially boils down to a crystalline solid that loses methanol solvate molecules, and as a result transforms to a different compound. Then on exposure to methanol in the gas or liquid phase it transforms back. There are innumerable examples of this, and two decades ago authors tried to claim this kind of "reversibility" as egress and ingress into permanent pores, but it is simply desolvation with collapse of the network and then nucleation and growth to the re-solvated phase. The authors have properly described it as such, but this is nothing new. The authors describe the methanol solvated phase as "metastable" but that is an improper use of that term here. Phases can be metastable or stable only if the composition remains constant.

Also, the notion that methanol solvent release is a proxy for methane release from gas hydrates is absurd. This idea would then be applicable to any crystal with a methanol solvate, of which there are thousands. And it is widely known that small molecule solvent molecules are readily lost under ambient conditions from many kinds of frameworks, including MOFs and hydrogen-bonded frameworks, as well as crystalline solids.

Reviewer #2 (Remarks to the Author):

The article is interesting since it describes the application of an ionic hydrogen-bonded organic framework (i-HOF) in reversible and automatic methanol (MeOH) adsorption/release. Conceptually, Gd-B mimics the NGH behavior by decomposing and release all MeOH molecules at ambient environment, while the structure could be recovered by simply exposing back to MeOH vapor or liquid. However, from my point of view, additional experiments/characterization studies, as well as concept clarifications, would be needed to validate this approach as carrier for alternative energy source. In the current form, the article would not be publishable in Nat. Commun., suggestions are enclosed.

1. The authors claim that Gd-B mimics natural gas hydrate (combustible ice) behavior at ambient condition. However, for the natural gas hydrate, it needs harsh condition due to the sorbate is gas molecules (primarily methane). Necessary clarification should be provided in the article for this conception.

2. The authors mention that the significance of MeOH adsorption comes from the importance of MeOH storage as a liquid fuel in industry. However, it doesn't seem to make that much sense to trap MeOH, a liquid fuel under ambient environment, inside iHOFs.

3. For the releasing of MeOH from Gd-B, it shows that only exposure to air would induce this transformation. To provide evidences, it would be necessary to incorporate additional data related with the influence of other gases, for example, nitrogen and carbon dioxide. In addition, vacuum level for the 24h vacuum drying process should be provided here.

4. Since the material was first reported by another group, the authors in this manuscript should include more information about how they contribute to this application. For example, based on

what mechanisms the non-porous hydrogen-bonded borate networks in the initial paper could be considered as a good sorbent for MeOH.

5. In terms of the adsorption capability, it would be necessary to establish a comparison with other MeOH-sorbents.

6. What is the stability of the pristine Gd-B material? Related experiments should be included.

Reviewer #3 (Remarks to the Author):

The manuscript described an ionic hydrogen-bonded organic framework (air-dried Gd-B) that captures methanol vapor through structural transformation. The framework can uptake the vapor after regeneration under mild activation condition and this work has a high novelty because there are few similar reported works among related studies using hydrogen-bonded organic frameworks. However, the additional experimental data and explanation must be added to improve quality. Hence, the work is acceptable for Nature Communications after the authors modify following points.

- To understand the number of guest molecules or regeneration condition with thermal stability of the framework, thermogravimetric data of two samples (fresh and air-dried ones) should be included in the manuscript with brief descriptions including the amount of guest molecules in the framework and structural decomposition.

- The authors only investigated porosity of the fresh Gd-B sample. However, porosity and other gas sorption properties of air-dried Gd-B is more important to compare with the data of the fresh Gd-B. So, these data should be added in the manuscript.

- In supplementary figure 5, PXRD pattern of air-dried Gd-B is similar but not identical with that of guanidinium tetraborate dihydrate (Especially, the peaks over 15° are so different). So, I cannot anticipate the structural arrangement of boron and guanidinium moieties. Through Rietveld refinement from the patterns, accurate structural information of air-dried Gd-B could be investigated and enhance the quality of this work.

- Theoretical and experimental element analysis values are not matched (the difference > 0.4%) in table S2, indicating existence of impurity or miscalculation of the formula of sample. This point should be clearly addressed. The author should add both elemental analysis data of fresh Gd-B and air-dried Gd-B with additional explanation.

- In manuscript, the authors just showed release of adsorbed methanol by irradiation of lighting in the video. There are short explanations for experimental process and characterization process. The authors should investigate the amount of released methanol and described detailed information of light source.

Response to Reviewers' comments

Reviewer #1 (Remarks to the Author):

This manuscript is far below the standards of Nature Communications. The behavior described is neither novel, unique, or interesting. There is a growing trend to use the term HOF. The authors here have now expanded on this to iHOF, i meaning ionic. Some authors (but not these) have taken to numbering them, like HOF-10, HOF-11, ignoring hundreds of "HOFs" reported long before. The use of this acronym, in the view of this reviewer, is an unsavory practice and it ignores a lot of history.

Response: The authors appreciate the comments from Reviewer 1.

Actually, the concept and use of this acronym of "iHOF" or "charge-assisted HOF" are not initialized by our group. Instead, exactly like what you said, the recent reviews and research articles reported some iHOF or charge-assisted HOFs (*Chem.Soc.Rev.*, **2019**, 48, 1362; *Acc. Chem. Res.*, **2016**, 49, 2669-2679; *CrystEngComm*, **2018**, 20, 1779; *Acc. Chem. Res.*, **1990**, 23, 120-126; etc.), including the HOF-GS-10 and HOF-GS-11 materials (*Angew. Chem.*, **2016**, 128, 10825-10829) that you mentioned.

We don't intend to seek popularity by denoting the notation of iHOF, which is simply used for convenience and to make difference from HOFs assembled by organic molecules. A brief history of HOF development has already been discussed in main text (pages 4 and 5, lines 69-87) and related references have been cited (references 14-32).

This article essentially boils down to a crystalline solid that loses methanol solvate molecules, and as a result transforms to a different compound. Then on exposure to methanol in the gas or liquid phase it transforms back. There are innumerable examples of this, and two decades ago authors tried to claim this kind of "reversibility" as egress and ingress into permanent pores, but it is simply desolvation with collapse of the network and then nucleation and growth to the re-solvated phase. The authors have properly described it as such, but this is nothing new.

Response: The reviewer has summarized the work accurately in short sentence, but ignore the conditions. Nevertheless, if simply casting a veil over the condition, water can be directly thermolyzed into H₂ and O₂ but at the temperature > 2000 °C, what sense it makes? The critical point in our work relies on that the reversible methanol adsorption-release take place *at the ambient conditions without external energy input*. To the best of our knowledge, we don't find other similar examples in literature.

The authors describe the methanol solvated phase as "metastable" but that is an improper

use of that term here. Phases can be metastable or stable only if the composition remains constant.

Response: “Metastable” means a system having or characterized by only a slight margin of stability. “Metastable phase” can be transformed from one structure to another structure while the composition remains constant. We don’t think that the word of “metastable” itself is related with the concept of phase and composition. In literature, some MOFs are also described as metastable MOF, in which the composition is changed. (See examples in publications: ACS Cent. Sci. 2015, 1, 252–260; J. Am. Chem. Soc. 2015, 137, 49, 15406)

Also, the notion that methanol solvent release is a proxy for methane release from gas hydrates is absurd.

Response: We can’t understand how the reviewer comes to such conclusion that “methanol solvent release is a proxy for methane release from gas hydrates”. This is an obvious misunderstanding or misrepresentation. In the context, we are talking about Gd-B release methanol solvent upon the Gd-B skeleton collapsing. This release behavior is similar with methane release from gas hydrates upon water cage structure breaking down. That is what we call “combustible ice mimicking”, rather than methanol release instead of methane release. For us, the reviewer is delivering misleading and arbitrary inference via interpreting out of context.

This idea would then be applicable to any crystal with a methanol solvate, of which there are thousands. And it is widely known that small molecule solvent molecules are readily lost under ambient conditions from many kinds of frameworks, including MOFs and hydrogen-bonded frameworks, as well as crystalline solids.

Response: In principle, adsorption is a spontaneous process with Gibbs energy $\Delta G < 0$, whereas desorption process requires external energy input, for example heating and/or dynamic vacuum (reduced pressure), as displayed in Fig. 1 in manuscript.

Adsorbed permanent gases (N_2 , Ar, CO_2 , CH_4 etc.) with boiling points much lower than room temperature are readily lost at ambient conditions. The release of volatile organic compounds (VOCs) and water in pores of porous frameworks typically requires harsh conditions. This is why most porous adsorbents need to be activated by heating and/or dynamic vacuum treatment before sorption experiments. The activation conditions of some typical frameworks synthesized in MeOH and/or water are listed in Table R1, where heating and reduced pressure are required to release the solvents accommodated inside. The release and adsorption of MeOH of Gd-B, accompanied with structure collapse and recovery, can be realized at ambient condition without external energy input, thus distinguishing it from most reported adsorbents.

Table R1 Activation conditions for small molecule release in various adsorbents.

Frameworks	Activation condition	Adsorbate	Ref
Cu(mtpm)Cl ₂ ·20H ₂ O	outgassed at 100 °C for 24 h	methanol, ethanol, water	[R1]
M-VNU-74-1	room temperature for 24 h under reduced pressure (20 mTorr), followed by evacuation at 70 °C for 12 h	methanol	[R2]
Gu(GLA((4,4'-Bipy) _{0.5}	150 °C overnight under vacuum	methanol, water	[R3]
(H ₂ dab)[Zn ₂ (ox) ₃]·6H ₂ O	80 °C/12 h under vacuum	water, ethanol and methanol	[R4]
JUC-110	160 °C/17 h under vacuum	ethanol, water	[R5]
Cu ₂ (pzdc) ₂ (dpyg)	100 °C/5 h under high vacuum	methanol, water	[R6]
MIL-101(Cr)	30-300 °C for 12 h under high vacuum (<10 ⁻⁵ torr)	water	[R7]
H ₂ O@iso1	100 °C/16h under vacuum	methanol, water	[R8]

[R1] Shigematsu, A., Yamada, T. & Kitagawa, H. Selective separation of water, methanol, and ethanol by a porous coordination polymer built with a flexible tetrahedral ligand. *J. Am. Chem. Soc.* **134**, 13145-13147 (2012).

[R2] Nguyen, B. T. et al. High methanol uptake capacity in two new series of metal-organic frameworks: promising materials for adsorption-driven heat pump applications. *Chem. Mater.* **28**, 6243-6249 (2016).

[R3] Chen, B. et al. Metal-organic framework with rationally tuned micropores for selective adsorption of water over methanol. *Inorg. Chem.* **47**, 5543-5545 (2008).

[R4] Sadakiyo, M., Yamada, T. & Kitagawa, H. Hydroxyl group recognition by hydrogen-bonding donor and acceptor sites embedded in a layered metal-organic framework. *J. Am. Chem. Soc.* **133**, 11050-11053 (2011).

[R5] Borjigin, T. et al. A microporous metal-organic framework with high stability for GC separation of alcohols from water. *Chem. Commun.* **48**, 7613-7615 (2012).

[R6] Kitaura, R. et al. A pillared-layer coordination polymer network displaying hysteretic sorption: $[\text{Cu}_2(\text{pzdc})_2(\text{dpyg})]_n$ (pzdc=pyrazine-2,3-dicarboxylate; dpyg=1,2-Di(4-pyridyl)glycol). *Angew. Chem.* **114**, 141-143 (2002).

[R7] Seo, Y. K. et al. Energy-efficient dehumidification over hierarchically porous metal-organic frameworks as advanced water adsorbents. *Adv. Mater.* **24**, 806-810 (2012).

[R8] Ferrando-Soria, J. Selective gas and vapor sorption and magnetic sensing by an isorecticular mixed-metal-organic framework. *J. Am. Chem. Soc.* **134**, 15301-15304 (2012).

Reviewer #2 (Remarks to the Author):

The article is interesting since it describes the application of an ionic hydrogen-bonded organic framework (i-HOF) in reversible and automatic methanol (MeOH) adsorption/release. Conceptually, Gd-B mimics the NGH behavior by decomposing and release all MeOH molecules at ambient environment, while the structure could be recovered by simply exposing back to MeOH vapor or liquid. However, from my point of view, additional experiments/characterization studies, as well as concept clarifications, would be needed to validate this approach as carrier for alternative energy source. In the current form, the article would not be publishable in Nat. Commun., suggestions are enclosed.

Response: The authors appreciate the valuable comments from Reviewer 2.

1. The authors claim that Gd-B mimics natural gas hydrate (combustible ice) behavior at ambient condition. However, for the natural gas hydrate, it needs harsh condition due to the sorbate is gas molecules (primarily methane). Necessary clarification should be provided in the article for this conception.

Response: Generally, natural gas hydrate features three structural models (*Nature*, **2003**, 426, 353-35). The formation of water cage and inclusion of guest molecules (primarily methane) typically requires low temperature and/or high pressure. The structure collapse accompanied with release of guest molecules would take place upon exposing to ambient condition. In this work, we use the term “mimicking” to emphasize that Gd-B can play similar behaviors as natural gas hydrate, that is, the adsorption and release of guest molecules with host structure collapse and re-building. While the main difference lies in the

fact that the structure collapse and restoration of Gd-B can be realized at ambient condition without external energy input, accompanied with automatic methanol release/adsorption. In summary, the term “mimicking” refers to the similarity between natural gas hydrate and Gd-B in terms of their release/adsorption behaviors, while the operation conditions and adsorbates are different. According to your advice, we have modified the statements to make clarification clear (page 5, lines 87-91).

2. The authors mention that the significance of MeOH adsorption comes from the importance of MeOH storage as a liquid fuel in industry. However, it doesn't seem to make that much sense to trap MeOH, a liquid fuel under ambient environment, inside iHOFs.

Response: Previously we mentioned that MeOH adsorption can be used in direct MeOH fuel cell. There are also reports on metal-organic frameworks in terms of their methanol adsorption for adsorption-driven heat pumps (*Chem. Rev.*, **2015**, 115, 12205-12250). Taking your comment into consideration, here we rephrase the relevant statement and aim to emphasize the concept developed in this work and its future potential (page 10, lines 176-182). That is, the adsorption/release proceeded at ambient condition without external energy input, can be regarded as a guidance for rational design of functional framework-based materials as carriers for alternative energy source.

3. For the releasing of MeOH from Gd-B, it shows that only exposure to air would induce this transformation. To provide evidences, it would be necessary to incorporate additional data related with the influence of other gases, for example, nitrogen and carbon dioxide. In addition, vacuum level for the 24h vacuum drying process should be provided here.

Response: As described in the main text, this transformation includes two processes, MeOH release (desorption) and hydrolysis of tetra-methyl borate when Gd-B was exposed to moisture in air. However, the hydrolysis will not take place when Gd-B was under vacuum, upon which only guest MeOH molecules can be released, as evidenced by the XRD (Supplementary Fig. 6) and MeOH sorption results (Fig. 5). There Gd-B is stable in water-free environment and structural transformation will not occur. As for the influence of other gases on Gd-B structure, our campus and lab shut down due to the COVID-19 pandemic, the tests are unavailable at this moment. According to the N₂ sorption isotherm experiment where Gd-B is exposed to N₂ at varied pressure, we believe that inert gas will not affect the Gd-B framework structure. The information has been added in the main text (page 9, lines 159-161, 169-171).

In addition, the pressure can reach down to 10e-7 Pa after 30 min vacuum using turbo molecular pump affiliated to BEL sorption instrument, which keeps stable with prolonging

vacuum time. The data has now been provided in Supplementary Methods.

4. Since the material was first reported by another group, the authors in this manuscript should include more information about how they contribute to this application. For example, based on what mechanisms the non-porous hydrogen-bonded borate networks in the initial paper could be considered as a good sorbent for MeOH.

Response: In 2005, Richard Robson and coauthors reported the synthetic route and crystal structure of Gd-B as the first generation of HOF comprised of cations and anions. (*J. Am. Chem. Soc.*, **2005**, 127, 816-817). In the paper, authors did not measure the porosity and sorption-related properties of Gd-B. Our contribution is that we found Gd-B can realize structural collapse and restoration upon automatic release and adsorption of MeOH at ambient condition, without external energy input. All these information have been fully discussed in the manuscript. We have to mention that the previous report of Gd-B crystal structure will exert little effect on the novelty of our work, as many structures have been reported firstly and then hundreds of papers are published to explore their new properties.

5. In terms of the adsorption capability, it would be necessary to establish a comparison with other MeOH-sorbents.

Response: Here we list the adsorption capability of various MeOH-adsorbents in Table R2 for reference. It is obvious that Gd-B shows very high MeOH adsorption capability among the reported adsorbents, which is comparable with MIL-100 having specific area > 3000 m²/g. The information has been added into the revised manuscript (page 10, lines186-188, and Supplementary Table 3).

Table R2 Comparison of MeOH adsorption capability among various adsorbents.

Material	Adsorption amount (g/g)	Ref
Gd-B*	0.60 (417 cm ³ /g)	this work
UiO-67	0.34	[R9]
ZIF-8(Zn)	0.37	[R10]
MIL-53(Cr)	0.53	[R11]
MIL-100(Cr)	0.67	[R12]
HKUST-1	0.50	[R13]
Al(OH)-(1,4-NDC)	0.16	[R14]
Zn ₂ (bptc)	0.10	[R15]

*The adsorption amount of air-dried sample includes the physical adsorption of MeOH and MeOH adsorption for the formation of borate to re-construct Gd-B framework.

[R9] Katz, M. J., et al. A facile synthesis of UiO-66, UiO-67 and their derivatives. *Chem. Commun.* **49**, 9449-9451 (2013).

[R10] Park, K. S., et al. Exceptional chemical and thermal stability of zeolitic imidazolate frameworks. *Proc. Natl. Acad. Sci. U. S. A.* **103**, 10186-10191 (2006).

[R11] Borelly, S., et al. Exploration of the adsorption of polar vapors in the highly flexible metal organic framework MIL-53(Cr). *J. Am. Chem. Soc.* **132**, 9488-9498 (2010).

[R12] Férey, G., et al. A hybrid solid with giant pores prepared by a combination of targeted chemistry, simulation, and powder diffraction. *Angew. Chem., Int. Ed.* **43**, 6296-6301 (2004).

[R13] Jeremias, F., et al. Water and methanol adsorption on MOFs for cycling heat transformation processes. *New J. Chem.* **38**, 1846-1852 (2014).

[R14] Comotti, A., et al. Nanochannels of two distinct cross-sections in a porous Al-based coordination polymer. *J. Am. Chem. Soc.* **130**, 13664-13672 (2008).

[R15] Lin, X., et al. A porous framework polymer based on a zinc(II) 4,4'-bipyridine-2,6,2',6'-tetracarboxylate: synthesis, structure, and "zeolite-like" behaviors. *J. Am. Chem. Soc.* **128**, 10745-10753 (2006).

6. What is the stability of the pristine Gd-B material? Related experiments should be included.

Response: The stability of pristine Gd-B was not mentioned when it was firstly reported in 2005 (*J. Am. Chem. Soc.*, **2005**, 127, 816-817). We found that the single-crystal obtained from the mother solution (solvent: mixture of MeOH and TEA) can retain its structure after rapid washing for three times using MeOH and exposing in air atmosphere for 0.5 h. However, the structure collapsed and finally transformed to air-dried Gd-B with prolonging exposure time (Supplementary Fig. 2). It is worthy to note that fresh Gd-B is structurally stable under vacuum (Supplementary Fig. 6). The data have already been included in the manuscript.

Reviewer #3 (Remarks to the Author):

The manuscript described an ionic hydrogen-bonded organic framework (air-dried Gd-B) that captures methanol vapor through structural transformation. The framework can uptake

the vapor after regeneration under mild activation condition and this work has a high novelty because there are few similar reported works among related studies using hydrogen-bonded organic frameworks. However, the additional experimental data and explanation must be added to improve quality. Hence, the work is acceptable for Nature Communications after the authors modify following points.

Response: The authors thank very much for the positive comments from Reviewer 3. Below please find the detailed responses.

- To understand the number of guest molecules or regeneration condition with thermal stability of the framework, thermogravimetric data of two samples (fresh and air-dried ones) should be included in the manuscript with brief descriptions including the amount of guest molecules in the framework and structural decomposition.

Fig. R1 Thermogravimetric curves of fresh Gd-B and air-dried Gd-B recorded from 25 to 800 °C at a heating rate of 10 °C/min in N₂ atmosphere.

Response: As shown in Supplementary Fig. 2, fresh Gd-B can only retain its structure within 0.5 h when exposing in air atmosphere. Prolonging exposure time leads to the transformation into air-dried Gd-B, which is stable in ambient condition. In addition, both fresh Gd-B and air-dried Gd-B are temperature-sensitive and not stable at elevated temperature, as indicated by the thermogravimetric data shown below (Fig. R1). Therefore, it is difficult to precisely measure the weight loss from guest molecule release in the framework. In this regard, MeOH adsorption/desorption tests at constant temperature can provide more convincing evidence to reveal the number/amount of guest molecules and transformation process of Gd-B, as they are recorded at room temperature and in line with the transformation conditions of fresh Gd-B and air-dried Gd-B.

As shown in Fig. 5b in main text, the MeOH sorption isotherms over air-dried Gd-B at 298 K (five runs) indicate that the complete recovery of Gd-B structure from air-dried Gd-B sample includes two main procedures: (1) chemical adsorption of twelve MeOH for the formation of borate ester; (2) physical adsorption of four MeOH as guest molecules within the framework. The detailed adsorption/desorption amounts can be found in the main text, which keep good consistency with our analyses. After the fifth-run of MeOH adsorption test, the Gd-B framework was recovered, as the sorption isotherm of air-dried Gd-B is overlapped with that of fresh Gd-B which physically adsorbed four MeOH guest molecules (Fig. 5c in main text).

- The authors only investigated porosity of the fresh Gd-B sample. However, porosity and other gas sorption properties of air-dried Gd-B is more important to compare with the data of the fresh Gd-B. So, these data should be added in the manuscript.

Response: As described in the manuscript, the framework of fresh Gd-B collapsed when exposed in air and finally transformed to air-dried Gd-B. We have evaluated the nitrogen sorption isotherms of air-dried Gd-B (Fig. R2), which shows no porosity. Now the data has been added into Supplementary Information (Supplementary Fig. 9).

Fig. R2 N₂ sorption isotherms of fresh Gd-B and air-dried Gd-B recorded at 77 K.

In supplementary Figure 5, PXRD pattern of air-dried Gd-B is similar but not identical with that of guanidinium tetraborate dihydrate (Especially, the peaks over 15° are so different). So, I cannot anticipate the structural arrangement of boron and guanidium moieties. Through Rietveld refinement from the patterns, accurate structural information of air-dried Gd-B could be investigated and enhance the quality of this work.

Response: As described in Supplementary Methods, the single crystal of guanidinium tetraborate dihydrate was obtained by recrystallization of air-dried Gd-B aqueous solution at room temperature. Under such condition, no further reaction can be expected. Therefore, we proposed that the structural difference between guanidinium tetraborate dihydrate and air-dried Gd-B originates from the water solvation effect (water molecules existing in the crystal lattice of guanidinium tetraborate dihydrate). Surely, accurate structural information of air-dried Gd-B could enhance the quality of this work through Rietveld refinement. Unfortunately, owing to the CoVID-19 epidemic, synchrotron radiation experiments can't be performed currently. According to the current data, we describe structure of air-dried Gd-B comprised of borate and guanidium as shown in Fig. 4c. On the other hand, the air-dried Gd-B structure will not seriously affect the integrity of the work. We are thankful for your kind understanding under this special situation.

- Theoretical and experimental element analysis values are not matched (the difference > 0.4%) in table S2, indicating existence of impurity or miscalculation of the formula of sample. This point should be clearly addressed. The author should add both elemental analysis data of fresh Gd-B and air-dried Gd-B with additional explanation.

Response: The elemental analyses are carried out on The Vario EL III Elemental Analyzer with an instrumental error of $\pm 0.3\%$. Therefore, the difference ca. 0.4% is somehow acceptable. The deviation of experimental data of element analysis for fresh Gd-B from theoretical value is probably due in large part to the release of MeOH during test.

- In manuscript, the authors just showed release of adsorbed methanol by irradiation of lighting in the video. There are short explanations for experimental process and characterization process. The authors should investigate the amount of released methanol and described detailed information of light source.

Response: The experimental process and characterization process can be described as follows: **Automatic release of four guest MeOH molecules from bulky Gd-B.** Typically, fresh Gd-B stored in MeOH was collected and put onto filter paper, upon which the surface attached MeOH on Gd-B were removed. Subsequently, the fresh Gd-B crystal (500 mg) was placed into a silica tube (length: 15 cm, inner diameter: 10 mm) and covered by sealing film. The tube was left untouched for 30 min to be fully filled with released MeOH guest molecules. An ordinary lighter was then placed at the tube outlet to ignite the released guest MeOH molecules from bulky Gd-B, the flame can last for few seconds (Supplementary Video 1). The amount of the released methanol has been determined by methanol sorption isotherm as mentioned above (four physically adsorbed methanol molecules in each unit cell, 15.8 wt%). The stability has been fully discussed in the main text. The information has been added into the Supplementary information.

REVIEWERS' COMMENTS:

Reviewer #2 (Remarks to the Author):

The authors have satisfactorily addressed all the comments from the reviewer and the paper can now be accepted as is.

Reviewer #3 (Remarks to the Author):

The revised manuscript is improved and I recommend acceptance of this paper.

Point-by-point responses to the referees' comments

(Referees' comments and the responses are displayed in black and blue, respectively)

Reviewer #2 (Remarks to the Author):

The authors have satisfactorily addressed all the comments from the reviewer and the paper can now be accepted as is.

Response: Thanks a lot for your time and effort on our manuscript.

Reviewer #3 (Remarks to the Author):

The revised manuscript is improved and I recommend acceptance of this paper.

Response: We truly appreciate your time and great effort on our manuscript.